Identifying liver metastasis-related hub genes in breast cancer and characterizing SPARCL1 as a potential prognostic biomarker

Chen Mingkuan
Zheng Wenfang
Fang Lin fanglin2017@126.com
Tongji University School of Medicine, Department of Thyroid and Breast Division of General Surgery Shanghai Tenth People’s Hospital , Shanghai , Jing’an District , China
Agarwal Supreet
Electronic publication date: 2023 May 8
Publication date: 2023
Volume: 11
Electronic Location ID: e15311
Received 2022 Nov 23; Accepted 2023 Apr 6
Copyright: ©2023 Chen et al.
Copyright year: 2023
Copyright holder: Chen et al.
License: This is an open access article distributed under the terms of the Creative Commons Attribution License, which permits unrestricted use, distribution, reproduction and adaptation in any medium and for any purpose provided that it is properly attributed. For attribution, the original author(s), title, publication source (PeerJ) and either DOI or URL of the article must be cited.
License URL: https://creativecommons.org/licenses/by/4.0/

Keywords: Breast cancer, Liver metastasis, Biomarker, SPARCL1

Funding: The Shanghai Municipal Health Commission, China 202040157 The National Natural Science Foundation of China 82073204 This work was supported by the Shanghai Municipal Health Commission, China (Grant number 202040157) and the National Natural Science Foundation of China (Grant number 82073204). There was no additional external funding received for this study. The funders had no role in study design, data collection and analysis, decision to publish, or preparation of the manuscript.

==============================
Background

The liver is the third most common metastatic site for advanced breast cancer (BC), and liver metastases predict poor prognoses. However, the characteristic biomarkers of BC liver metastases and the biological role of secreted protein acidic and rich in cysteine-like 1 (SPARCL1) in BC remain unclear. The present study aimed to identify potential biomarkers for liver metastasis of BC and to investigate the effect of SPARCL1 on BC.

Methods

The publicly available GSE124648 dataset was used to identify differentially expressed genes (DEGs) between BC and liver metastases. Gene Ontology (GO) and Kyoto Encyclopedia of Genes and Genomes (KEGG) enrichment analyses were conducted to annotate these DEGs and understand the biological functions in which they are involved. A protein–protein interaction (PPI) network was constructed to identify metastasis-related hub genes and further validated in a second independent dataset (GSE58708). Clinicopathological correlation of hub gene expression in patients with BC was determined. Gene set enrichment analysis (GSEA) was performed to explore DEG-related signaling pathways. SPARCL1 expression in BC tissues and cell lines was verified by RT-qPCR. Further in vitro experiments were performed to investigate the biological functions of SPARCL1 in BC cells.

Results

We identified 332 liver metastasis-related DEGs from GSE124648 and 30 hub genes, including SPARCL1, from the PPI network. GO and KEGG enrichment analyses of liver-metastasis-related DEGs revealed several enriched terms associated with the extracellular matrix and pathways in cancer. Clinicopathological correlation analysis of SPARCL1 revealed that its expression in BC was associated with age, TNM stage, estrogen receptor status, progesterone receptor status, histological type, molecular type, and living status of patients. GSEA results suggested that low SPARCL1 expression in BC was related to the cell cycle, DNA replication, oxidative phosphorylation, and homologous recombination. Lower expression levels of SPARCL1 were detected in BC tissues compared to adjacent tissues. The in vitro experiments showed that SPARCL1 knockdown significantly increased the proliferation and migration of BC cells, whereas the proliferation and migration were suppressed after elevating the expression of SPARCL1.

Conclusion

We identified SPARCL1 as a tumor suppressor in BC, which shows potential as a target for BC and liver metastasis therapy and diagnosis.

Introduction

Breast cancer (BC) in women has now become the most newly diagnosed malignant tumor worldwide, with both the number of new cases and number of deaths ranking first among all malignant tumors (Sung et al., 2021). Generally, patients with BC have a relatively good prognosis, with approximately 44% of patients with early-stage BC having a nearly 100% 5-year survival rate, whereas once organ metastasis occurs, this survival rate drops sharply to 26% (Miller et al., 2019). The organs most likely to metastasize from BC are the lungs, bone, liver, and brain (Cummings et al., 2014). Metastasis of the liver can cause various fatal complications, including liver failure, intractable ascites, portal vein thrombosis, and malnutrition (Diamond, Finlayson & Borges, 2009). Thus, occurrence of liver metastases indicates a worse prognosis in patients with BC, with a reported median survival time of approximately 3 years (Zhao et al., 2018a), However, the molecular mechanism of metastasis remains unclear. Identifying potential molecular biomarkers of liver metastasis may provide more accurate information to guide clinical decisions and predict prognosis, as well as provide direction and theoretical support for future research on the mechanism of metastasis. Therefore, identifying DEGs with prognostic significance in BC liver metastases, determining their biological function and evaluating their potential as therapeutic targets remain essential.

The Gene Expression Omnibus (GEO) and The Cancer Genome Atlas (TCGA) databases have collected and stored a large amount of tumor sequencing data, which are publicly available. Re-analyzing these sequencing data using bioinformatics methods and mining for differences in genetic information among different samples can help provide a scientific explanation of the occurrence and progression of diseases (Gauthier et al., 2019). Therefore, based on public database and bioinformatics analyses, we screened differentially expressed genes (DEGs) between BC and liver metastasis, and identified hub genes.

SPARCL1, a member of the secreted protein acidic and rich in cysteine (SPARC) family, is an extracellular matrix (ECM) glycoprotein with a gene located at 4q22. SPARCL1 has been reported to be widely expressed in normal brain, lung, heart, colon, and muscle tissues (Hu et al., 2012a). It is associated with the regulation of cell proliferation and migration through the regulation of embryogenesis, matrix remodeling and tumorigenesis (Liu et al., 2021). Many studies have reported its suppressive role in the development of various malignancies. However, it has been less studied in BC, especially in BC liver metastasis, highlighting the need to further investigate the role of SPARCL1 in BC and liver metastasis.

In the present study, we first screened the BC liver metastasis-related genes via integrated bioinformatics analysis and identified SPARCL1 as a key gene. Then the expression of SPARCL1 and its correlation with the clinical characteristics of BC patients were analyzed. In addition, the biological function of SPARCL1 in BC was evaluated in vitro. In summary, we aimed to identify potential biomarkers of BC liver metastasis and assess the expression pattern and role of SPARCL1 in BC. These findings are expected to provide a new understanding of BC and liver metastasis, driving future research and providing potential biomarkers for BC diagnosis and therapy.

Materials & Methods

Discovery and validation datasets

In the GEO (http://www.ncbi.nlm.nih.gov/geo/), datasets GSE124648 (Sinn et al., 2019) and GSE58708 (McBryan et al., 2015) were obtained with the following keywords: breast neoplasms, breast cancer, liver metastasis, expression profiling by array, attribute name tissue, and Homo sapiens. Array data of liver metastases (N = 16) and primary tumors (control, N = 130) from the GPL96 (HG-U133A; Affymetrix Human Genome U133A Array) platform were selected for differential expression analysis. The array data for GSE58708 consisted of three patients with BC liver metastasis vs. three controls as a validation dataset (Table 1).

Table 1 Discovery and validation of breast cancer (BC) datasets.

Datasets	Platform	Experiment type	Number of cases (metastasis/primary)	
GSE124648	GPL96	Expression profiling by array	16/130	
GSE58708	GPL11154	Expression profiling by high-throughput sequencing	3/3	

DEG identification

R software (version 4.0.3; R Core Team, 2020) was employed for bioinformatics analysis. The “limma” package was adopted to identify DEGs between BC and liver metastasis (Ritchie et al., 2015), with the following cut-off criteria: —log2FC—>2.0 and adj.P.Val<0.05. Similarly, the R package “TCGAbiolinks” was used to obtain biological data for BC in the TCGA database (Colaprico et al., 2016), and hub gene expression was verified by analyzing the differentially expressed mRNAs in BC. “ggplot2” and “heatmap” packages were used for visualizing the DEGs.

Enrichment analysis of DEGs

The “clusterProfile” package was employed to conduct gene ontology (GO) and Kyoto Encyclopedia of Genes and Genomes (KEGG) enrichment analyses of the DEGs (Yu et al., 2012), and a gene set enrichment analysis (GSEA) method was used for the KEGG enrichment analysis. p < 0.05 was the criteria set for statistical significance.

Integration of PPI networks and identification of hub genes

The STRING (http://www.string-db.org/) database provides comprehensive information on protein–protein interactions (PPI) (Szklarczyk et al., 2021). A PPI network of DEGs was integrated using the STRING database. Then, the results were visualized and analyzed using Cytoscape (version 3.7.2). The important modules were extracted via the plug-in MCODE, with the cut-off criteria being an MCODE score>5 and nodes>5. The other default parameters were set as Max. Depth = 100, K-Core = 2, Node score cut-off = 0.2, Degree Cut-off = 2. The plug-in cytoHubba was employed to identify the top 30 hub genes in the PPI network, with the following cut-off criteria: degree.layout ≥25 (MCC algorithm) (Shannon et al., 2003). GEPIA (http://gepia.cancer-pku.cn/) was used to obtain the hub gene expression between BC and normal tissues, with the following default parameters: p ≤ 0.01 and —Log2FC— ≥1 (Tang et al., 2017). The Kaplan–Meier plotter (https://kmplot.com/analysis/) was used to evaluate the correlation between the expression of the hub genes and patient survival (Lanczky & Gyorffy, 2021). The median expression was used to divide the patients into high and low expression groups. Then, the Kaplan–Meier curves of overall survival (OS) and recurrence-free survival (RFS) were drawn. Finally, hub genes (SPARCL1 and SERPINA1) with prognostic values were identified. p < 0.05 using the log-rank test was considered statistically significant.

Validation of hub genes and clinicopathological correlation analysis

A box plot was plotted to verify SPARCL1 expression in liver metastasis based on the GSE58708 dataset. The correlations between SPARCL1 expression and clinicopathological characteristics of patients with BC were analyzed using TCGA-BRCA clinical data. Then, based on patient survival data and SPARCL1 levels, the optimal expression threshold was calculated using the “survminer” package. The correlation between gene expression and clinicopathological parameters was tested using Pearson’s chi-squared test. p < 0.05 was considered statistically significant.

GSEA

GSEA can be used to evaluate whether a predefined gene set shows statistically significant differences between two groups with different phenotypes. Expression (.gct) and phenotype (.cls) information files for SPARCL1 were uploaded to the GSEA software (version 4.1.0) to conduct enrichment analysis. The chip platform and gene set database selected were “Human_ENSEMBL_Gene_ID_MSigDB.v7.0” and “c2.cp.kegg.v7.1. symbols.gmt,” respectively. The normalized enrichment score was calculated. Both false discovery rate (FDR q-val) and nominal p-value (NOM p-val) < 0.05 were considered statistically significant.

Clinical specimens and cell lines

Tissue samples were obtained from patients with BC who underwent radical mastectomy at Shanghai Tenth People’s Hospital (Shanghai, China). A total of 30 pairs of tumor and matched normal adjacent breast tissues were collected and preserved in liquid nitrogen. All clinical samples were obtained with the written informed consent of participants, and the project was approved by the Ethics Committee of Shanghai Tenth People’s Hospital (No. 2020-KN174-01). BC cells (MDA-MB-231, BT549, and MCF-7) and mammary epithelial cells (MCF-10A) were purchased from the Cell Bank of Type Culture Collection of Chinese Academy and grown in mammary epithelial basal medium (Cambrex, East Rutherford, NJ, USA) and Gibco Dulbecco’s modified Eagle medium (DMEM; Thermo Fisher Scientific, Waltham, MA, USA) supplemented with 10% fetal bovine serum (FBS; Gibco) and 1% penicillin-streptomycin (Enpromise, Shanghai, China). All cells were maintained in a 5% CO2 incubator at 37 °C.

Cell transfection, RNA extraction, and RT-qPCR

Lentiviral vector pLKO.1 and lentiviral packaging kit (Genomeditech, Shanghai, China) was used for construct stable SPARCL1 knockdown cells. sh-NC and shRNAs targeting SPARCL1 (SPARCL1 sh-1 and SPARCL1 sh-2) were purchased from IBSbio (Shanghai, China) and the sequences were as follows: SPARCL1 sh-1: 5′- CCCACAATGATAACCAAGAAA-3′, SPARCL1 sh-2: 5′- GCAGAGAAATAAAGTCAAGAA-3′. The cells transduced by lentivirus were selected with 2 µg/ml puromycin for 3 days. The pcDNA3.1 vector was used to construct SPARCL1 overexpression plasmid (IBSbio, Shanghai, China). Plasmids were transfected into BC cells using Invitrogen Lipofectamine 3000 (Thermo Fisher Scientific).

A lentiviral short hairpin RNA (shRNA) construct allows for stable and long-term knockdown of the targeted gene. Invitrogen TRIzol reagent (Thermo Fisher Scientific) was used to extract total RNA from cell lines and tissues. HiScript III RT SuperMix for qPCR (Vazyme, Nanjing, China) was used to generate the cDNAs, and Hieff qPCR SYBR Green Master Mix (Yeasen, Shanghai, China) was used to conduct real-time quantitative-polymerase chain reaction (RT-qPCR) following the manufacturer’s protocol, with ACTB as internal control for normalizing SPARCL1 expression. We used the following primers to conduct RT-qPCR. SPARCL1 (forward: 5′-CCAACTGAAGGTACATTGGACAT-3′, reverse: 5′-CTGTGAAGGAACTAACACCAGG-3′) and ACTB (forward: 5′-CATGTACGTTGCTATCCAGGC-3′, reverse: 5′-CTCCTTAATGTCACGCACGAT-3′).

MTT, colony formation, wound healing, and Transwell assays

To evaluate the effects of SPARCL1 on BC cells, post-transfection BC cell lines were cultured in 96-well plates at a density of 1,500 cells. Next, 20 µL methylthiazolyldiphenyl-tetrazolium bromide (MTT; Yeasen) was added to each well at 0, 24, 48, 72, and 96 h after inoculation, and the samples were incubated for 4 h at 37 °C in a 5% CO2 incubator to assess cell viability. Then, 150 µL DMSO was added to each well after removing the supernatant. The absorbance was measured using a microplate spectrophotometer (BioTek Instruments, Winooski, VT, USA) at 490 nm. Cell proliferation curves were plotted according to the absorbance value.

Transfected BC cell lines were prepared as single-cell suspensions, inoculated at a density of 750 cells/well until prominent colonies were formed. The cells were fixed with 95% ethanol and stained with 0.1% crystal violet (Yeasen) to detect cell colony formation ability. Representative pictures were recorded, and clone colonies were counted.

To detect cell mobility, the cell monolayer was scratched with RNA free tips when the transfected BC cell density reached 90% or more. In the following step, DMEM supplemented with 2% FBS was used as the culture medium. The healing of scratches was observed at 0 and 12 h using the same field of view to calculate cell mobility.

The transfected BC cells and 500 µL of DMEM containing 10% FBS were added to Transwell’s upper and lower chambers (Corning, Corning, NY, USA), respectively. After culturing for 18 h, migrated cells were fixed with 4% paraformaldehyde and stained with 0.1% crystal violet to assess the migratory ability. Representative images were captured using an inverted microscope.

Statistical analysis

Wilcoxon matched-pairs signed-rank test was used to compare the expression of SPARCL1 between BC and control samples. Unpaired Student’s t-test was used to compare the expression of SPARCL1 between MCF-10A and BC cell lines. The results of the MTT assay were analyzed using a two-way analysis of variance. All experiments were repeated three times. The experimental data were analyzed and plotted using Prism v8.3.0 (GraphPad Software, San Diego, CA, USA). p < 0.05 was considered statistically significant.

Results

DEG identification

We identified 332 DEGs in the GSE124648 dataset, of which 116 and 216 were upregulated and downregulated genes, respectively (Fig. 1A). A heatmap was used to visualize the top 50 liver metastasis-related genes in the dataset (Fig. 1B). Among the top 50 DEGs, 60% showed significant down expression in liver metastasis compared to BC tissues. Another 40% of highly expressed DEGs showed higher expression compared to BC tissues. These results suggest that downregulated DEGs in BC liver metastasis probably play a more critical role than upregulated DEGs.

Figure 1 Differentially expressed genes (DEGs) of liver metastasis of breast cancer (BC) identified from GSE124648.

(A) Volcano plot of the 332 DEGs, where blue and red indicate downregulated and upregulated DEGs, respectively. The cut-off criteria were adj.P.Val <0.05 and —log2FC— >2.0. (B) Heatmap of the systematic cluster analysis of the top 50 DEGs.

GO and KEGG enrichment analysis of DEGs

We further performed functional and pathway enrichment analyses on the 332 DEGs. GO annotation divides gene function into three categories: cellular components (CC), molecular function (MF), and biological process (BP) (Sinn et al., 2019). The top eight enriched GO terms for each category are shown in Fig. 2. CC analysis indicated that these DEGs were particularly related to the extracellular matrix, endoplasmic reticulum lumen, collagen-containing extracellular matrix, collagen trimer, and blood microparticles (Fig. 2A). MF analysis showed that the DEGs were mainly involved in extracellular matrix structural constituents, glycosaminoglycan binding, extracellular matrix structural constituents conferring tensile strength, heparin-binding, and collagen binding (Fig. 2B). The BP category was mainly enriched in extracellular structure organization, extracellular matrix organization, wound healing, humoral immune response, and complement activation (Fig. 2C). The KEGG pathway enrichment analysis results further showed that the upregulated DEGs were significantly enriched in neutrophil extracellular trap formation, alcoholic liver disease, and neuroactive ligand-receptor interaction, whereas the downregulated DEGs were significantly enriched in malignancy-related pathways, including pathways in cancer, BC, PI3K-Akt signaling pathway, MAPK signaling pathway, and focal adhesion (Fig. 2D). The above results suggest that these DEGs were mainly enriched in entries associated with extracellular matrix remodeling, and remarkably, downregulated DEGs were significantly enriched in various signaling pathways involved in the regulation of malignant tumor pathogenesis.

Figure 2 Gene ontology (GO) annotation and Kyoto Encyclopedia of Genes and Genomes (KEGG) pathway enrichment analyses of the DEGs.

(A–C) GO annotation enrichment analyses of the 332 DEGs. CC, cellular component; MF, molecular function; BP, biological process. (D) KEGG pathway enrichment analyses of the DEGs, where blue and red indicate downregulated and upregulated DEGs, respectively.

PPI network and SPARCL1 identified as a prognostic-related hub gene

The PPI network contained 332 nodes and 2,972 edges (Fig. S1), and we selected the top three hub modules identified by the MCODE plug-in for display (Figs. 3A–3C), where each node represents one DEG and the edges between nodes represent interactions. The top 30 hub genes with a high degree of connectivity were identified from the network (Fig. 3D). The expression of these genes and their modules are listed in Table 2. Among the 30 hub genes, SERPINA1 and VCAN were found to be upregulated in BC, whereas ALB, IGFBP3, SPARCL1, and FSTL1 were downregulated and no significant difference was discovered in the expression of other hub genes. Survival analyses showed favorable OS and RFS in patients with BC with upregulated SERPINA1 and SPARCL1 expression (Fig. 4). However, it was paradoxical that SERPINA1 showed high expression in both breast cancer and liver metastasis, but predicted a better patient prognosis. Therefore, we selected SPARCL1 for further evaluation, as the above results also suggested that it was likely to be a diagnostic and prognostic biomarker for BC.

Figure 3 Identification of hub modules and genes from the protein–protein interaction (PPI) network.

(A–C) Top three modules with high scores identified from the PPI network. Blue and red indicate downregulated and upregulated DEGs, respectively. (D) Top 30 hub genes identified from the PPI network. Ordered from high (red) to low (yellow) connectivity degrees.

Table 2 Top 30 hub genes associated with BC liver metastasis.

Gene symbol	log2FC	adj.P.Val	Expression	Module	
FGG	6.149436	5.24 × 10−29	UP	1	
APOA1	4.69204	7. 81 × 10−21	UP	1	
IGFBP7	−2.9555	5.11 × 10−21	DOWN	1	
APOB	3.954259	1.62 × 10−25	UP	1	
FSTL1	−3.02757	2.78 × 10−28	DOWN	1	
PRSS23	−2.34018	1.94 × 10−10	DOWN	1	
VCAN	−2.54093	5.33 × 10−17	DOWN	1	
TNC	−2.71678	3.74 × 10−10	DOWN	1	
ORM1	6.685187	5.25 × 10−28	UP	2	
IGFBP3	−2.10444	1.15 × 10−15	DOWN	1	
LAMB1	−3.0713	8.26 × 10−22	DOWN	1	
SERPIND1	3.39324	4.63 × 10−13	UP	1	
SPP2	2.236803	4.46 × 10−15	UP	1	
IGFBP5	−2.35175	1.19 × 10−7	DOWN	1	
FGA	4.820105	1.17 × 10−28	UP	1	
GC	6.136118	8.01 × 10−31	UP	2	
FGB	5.56398	3.49 × 10−29	UP	2	
HRG	4.776554	1.28 × 10−25	UP	2	
CP	2.150198	2.51 × 10−5	UP	1	
FBN1	−4.03832	1.62 × 10−25	DOWN	1	
SERPINA10	2.515328	4.67 × 10−10	UP	1	
ALB	7.627917	3.85 × 10−34	UP	1	
APOA2	5.683622	9.91 × 10−27	UP	1	
AHSG	3.816468	2.15 × 10−19	UP	1	
SERPINA1	3.350588	1.20 × 10−13	UP	1	
SPARCL1	−3.75713	8.01 × 10−31	DOWN	1	
TF	4.63786	4.81 × 10−14	UP	1	
F5	2.037343	1.08 × 10−11	UP	1	
SERPINC1	4.646863	8.04 × 10−28	UP	1	
ITIH2	2.98753	6.57 × 10−26	UP	1	
Notes.

Abbreviationslog2FC log2(Fold Change)

adj.P.Val adjusted P value

Figure 4 mRNA expression and prognostic value of (A) SPARCL1 and (B) SERPINA1.

Red and black lines represent a patient with high and low gene expression, respectively. OS, overall survival; RFS, recurrence-free survival; HR, hazard ratio (* p < 0.05).

Validation of SPARCL1 and clinicopathological correlation analysis

To affirm the accuracy of the above results, we externally validated the independent dataset GSE58708. Following differential expression analysis of the GSE58708 dataset, 683 DEGs were obtained, including 410 upregulated and 273 downregulated genes (Fig. S2). SPARCL1 expression was also significantly downregulated in liver metastasis compared with that in BC tissues (log2FC = −2.617; Fig. S3A). In the TCGA–BRCA dataset, SPARCL1 expression in BC tissues was also significantly downregulated compared with that in normal mammary gland tissue (Fig. S3B). The correlation between SPARCL1 levels and the clinical characteristics of patients with BC was further analyzed and showed that the lower SPARCL1 expression was significantly related to age, tumor-node-metastasis (TNM) stage, estrogen receptor (ER) status, progesterone receptor (PR) status, histological type, molecular type, and living status, whereas no significant difference in node stage and Her-2 status was observed (Table 3). The above findings suggest that SPARCL1 expression was downregulated both in BC and liver metastasis, and its low expression correlates with younger age, high TNM stages, and HR negative status of patients, implying that SPARCL1 may play a tumor suppressive role in BC patients.

Table 3 Relationship between clinical characteristics of BC patients from the TCGA database and SPARCL1 expression level.

		SPARCL1 expression		
	Total
(N = 1,049)	High
(N = 432)	Low
(N = 617)	p-value	
Age (years)					
<55	432 (41.2%)	200 (46.3%)	232 (37.6%)	0.00592	
≥55	617 (58.8%)	232 (53.7%)	385 (62.4%)		
TNM stage					
I	175 (16.7%)	90 (20.8%)	85 (13.8%)	0.0027	
II	599 (57.1%)	221 (51.2%)	378 (61.3%)		
III	242 (23.1%)	110 (25.5%)	132 (21.4%)		
IV	20 (1.9%)	5 (1.2%)	15 (2.4%)		
Unknown	13 (1.2%)	6 (1.4%)	7 (1.1%)		
Node stage					
N0–N1	615 (58.6%)	245 (56.7%)	370 (60.0%)	0.139	
N2–N3	137 (13.1%)	51 (11.8%)	86 (13.9%)		
Unknown	297 (28.3%)	136 (31.5%)	161 (26.1%)		
ER status					
Negative	170 (16.2%)	31 (7.2%)	139 (22.5%)	<0.001	
Positive	571 (54.4%)	265 (61.3%)	306 (49.6%)		
Unknown	308 (29.4%)	136 (31.5%)	172 (27.9%)		
PR status					
Negative	238 (22.7%)	58 (13.4%)	180 (29.2%)	<0.001	
Positive	500 (47.7%)	235 (54.4%)	265 (42.9%)		
Unknown	311 (29.6%)	139 (32.2%)	172 (27.9%)		
HER2 status					
Negative	621 (59.2%)	251 (58.1%)	370 (60.0%)	0.145	
Positive	107 (10.2%)	37 (8.6%)	70 (11.3%)		
Unknown	321 (30.6%)	144 (33.3%)	177 (28.7%)		
Histological type					
IDC	752 (71.7%)	260 (60.2%)	492 (79.7%)	<0.001	
ILC	196 (18.7%)	137 (31.7%)	59 (9.6%)		
Others	100 (9.5%)	35 (8.1%)	65 (10.5%)		
Unknown	1 (0.1%)	0 (0%)	1 (0.2%)		
Molecular type					
Basal-like	94 (9.0%)	11 (2.5%)	83 (13.5%)	<0.001	
Lum A	219 (20.9%)	135 (31.3%)	84 (13.6%)		
Lum B	120 (11.4%)	21 (4.9%)	99 (16.0%)		
HER2-enriched	52 (5.0%)	14 (3.2%)	38 (6.2%)		
Normal-like	7 (0.7%)	5 (1.2%)	2 (0.3%)		
Unknown	557 (53.1%)	246 (56.9%)	311 (50.4%)		
Living status					
Alive	902 (86.0%)	385 (89.1%)	517 (83.8%)	0.0185	
Dead	147 (14.0%)	47 (10.9%)	100 (16.2%)		
Notes.

Abbreviations TCGA The Cancer Genome Atlas

TNM tumor-node-metastasis

ER estrogen receptor

PR progesterone receptor

HER2 human epidermal growth factor receptor 2

IDC Invasive ductal carcinoma

ILC invasive lobular carcinoma

Lum luminal

GSEA

From the viewpoint of enrichment of gene sets, finding the effects of subtle changes on biological pathways or functions is easier in theory (Subramanian et al., 2005). Therefore, we performed GSEA based on the different expression levels of SPARCL1 in BC. The results suggested that 50 gene sets were upregulated in the low expression SPARCL1 phenotype, 47 gene sets were significantly enriched at FDR <25%, and 40 gene sets were significantly enriched at nominal p-value <5%. Figure 5 displays six signaling pathways (DNA replication, cell cycle, oxidative phosphorylation, homologous recombination, spliceosome, and proteasome) that were significantly enriched when SPARCL1 was downregulated in BC, revealing the potential molecular mechanisms by which SPARCL1 participates in BC occurrence and progression.

Figure 5 Single-gene gene set enrichment analysis (GSEA) of low SPARCL1 expression in BC.

NES, normalized enrichment score; FDR, false discovery rate; NOM p-val, nominal p-value.

SPARCL1 downregulation in BC tissues and cells

Total RNA was extracted from tissues and cell lines for RT-qPCR to validate SPARCL1 expression in BC. In comparison with the paired adjacent normal breast tissues, SPARCL1 was significantly downregulated in BC tissues (N = 30; Fig. 6A). In comparison with the normal breast epithelial cell line, SPARCL1 expression was also decreased in the BC cell lines (Fig. 6B). These results are consistent with those from our bioinformatics analysis, suggesting that SPARCL1 is indeed expressed significantly less in BC compared to in normal breast tissue.

Figure 6 SPARCL1 expression in tissues and cell lines detected by RT-qPCR.

(A) SPARCL1 expression in BC tissues and matched normal adjacent breast tissues (Wilcoxon matched-pairs signed-rank test). (B) Expression of SPARCL1 in MCF-10A and breast cancer cell lines (Student’s t-test). * p < 0.05, ** p < 0.01, *** p < 0.001, **** p < 0.0001.

SPARCL1 knockdown-induces proliferation and migration of BC cells in vitro

To explore the biological function of SPARCL1, we used shRNAs to knockdown SPARCL1 in BC cells. The results suggest that shRNAs could significantly downregulate the level of SPARCL1 in BC cells compared to sh-NC (Fig. 7A). The results of colony formation and MTT assays suggested that SPARCL1 inhibition promoted the colony formation and proliferation of BC cells (Figs. 7B and 7C). Cell migration is an essential step in tumor progression and metastasis. In comparison with the control group (sh-NC), the healing ability of SPARCL1-knockdown MDA-MB-231 cells was significantly enhanced (Fig. 7D) and cell migration was significantly increased (Fig. 7E). Collectively, these results suggested that downregulation of SPARCL1 promoted the colony formation, proliferation, and migration of BC cells.

Figure 7 Biological function experiments in BC cells upon SPARCL1 knockdown.

(A) RT-qPCR assessing the knockdown efficiency of SPARCL1 shRNA on BC cells. (B) Colony formation assay measuring the proliferation ability of SPARCL1-knockdown BC cells. Cell colonies were photographed and their number was counted. (C) MTT assay was performed to detect the proliferation of SPARCL1-knockdown BC cells. (D) Wound healing assay was performed to assess the migration of MDA-MB-231 cells. Cells were photographed at 0 and 12 h after scratching, and wound sizes were compared. (E) The migration ability of MDA-MB-231 was evaluated with a Transwell assay, and the number of cells was counted. * p < 0.05, ** p < 0.01, *** p < 0.001, **** p < 0.0001. NC, normal control; sh, short hairpin RNA.

Overexpression of SPARCL1 inhibited the proliferation and migration of BC in vitro

To further evaluate the effects of overexpression of SPARCL1 on the biological function of BC cell lines, SPARCL1 overexpression plasmid was transfected into BC cells. The results of RT-qPCR showed that the expression of SPARCL1 was significantly upregulated in the SPARCL1 OE group compared to the control (vector) group (Fig. 8A). MTT assay revealed that the proliferation of BC cells was significantly decreased after the overexpression of SPARCL1 (Fig. 8C), and the results of the colony formation assay showed that overexpression of SPARCL1 inhibited the colony formation of BC cells (Fig. 8B). The results of the wound-healing assay showed that there was a significant decrease in cell migration area after SPARCL1 overexpression compared to the control group (Fig. 8D). The results of Transwell assay showed that the number of cells in the lower chamber was significantly reduced after SPARCL1 overexpression (Fig. 8E). These findings demonstrated that the colony formation, proliferation, and migration of BC cells were inhibited after elevating SPARCL1 expression.

Figure 8 Biological function experiments in BC cells upon SPARCL1 overexpression.

(A) RT-qPCR assessing the overexpression efficiency of SPARCL1 (B) Colony formation assay to detect the effect of overexpression of SPARCL1 on BC cells colony formation ability. (C) Cell proliferation was assessed in BC cells by MTT assay. (D) Wound healing assay was performed to assess the effect of overexpression of SPARCL1 on the migration ability of BC cells and the quantitative comparison of the percentage of scratch migration area. (E) Transwell assay was performed to detect the effect of overexpression of SPARCL1 on the migration ability of BC cells and the quantitative comparison of the number of migrating cells. * p < 0.05, ** p < 0.01, *** p < 0.001, **** p < 0.0001. OE, overexpression.

Discussion

Bioinformatics analysis of sequencing data can improve our understanding of gene function, including gene expression patterns between different experimental conditions or phenotypes, identification of biological processes related to gene expression, and screening of therapeutic targets and prognostic markers (Hu et al., 2015; Kaifi et al., 2015; Tao et al., 2017). In this study, 332 DEGs associated with liver metastasis were identified by differential expression analysis of sequencing data from publicly available datasets. GO enrichment analysis suggested that these DEGs were mostly located in the extracellular matrix and participated in biological processes, including extracellular matrix organization, wound healing, angiogenesis, and humoral immune response. Extracellular matrix organization is closely correlated with the occurrence and progression of cancer; a neatly arranged matrix can promote tumor cell invasion. The tumor suppressor PTEN participates in the regulation of matrix remodeling, which is negatively correlated with the arrangement of collagen in human breast tissue (Jones et al., 2019). Tumor angiogenesis can supply nutrients and oxygen essential for tumor growth and metastasis (Weis & Cheresh, 2011). Hypoxia-inducible factor-dependent angiogenesis is vital for the invasion, progression, and drug resistance of BC and is closely correlated with its poor prognosis (De Heer, Jalving & Harris, 2020). KEGG pathway analysis showed that downregulated DEGs were particularly enriched in malignant tumor-related pathways, such as MAPK signaling, PI3K/AKT signaling, focal adhesion, Wnt signaling, and Hippo signaling. The MAPK signaling pathway is crucial for BC invasion and metastasis, promoting the occurrence and progression of the disease (Cotrim et al., 2013; Jiang et al., 2020; Ke et al., 2021). In BC, more than 70% patients have PI3K signaling pathway alterations and its activation is significantly associated with an HR-negative, basal-like phenotype, high histological grade, and cancer-specific death (López-Knowles et al., 2010). The PI3K/AKT signaling pathway also participated in liver metastasis of various malignant tumors. Indeed, microRNA-582 can promote gastric cancer liver metastasis via the PI3K/AKT/Snail pathway mediated by FOXO-3 (Xie et al., 2020). The c-Met/PI3K/AKT/mTOR axis can activate the liver metastasis-specific cholesterol metabolism pathway in colorectal cancer, providing conditions for tumor cell colonization and growth in the liver (Zhang et al., 2021a). There are also reports on the influence of Wnt, Hippo, and other signaling pathways on the liver metastasis of malignant tumors (Chai et al., 2019; Yuan et al., 2019). Based on the above results, we suggested that these DEGs, especially those which are downregulated, alter the tumor microenvironment primarily through regulating the extracellular matrix, angiogenesis, and humoral immunity, and are involved in the mechanism of BC liver metastasis through multiple signaling pathways, which is moreover a multi-gene co-regulation process rather than a single independent gene or product.

Based on the prognostic value of hub DEGs in BC, we identified SPARCL1, a member of the secreted protein acidic and rich in cysteine (SPARC) family, which is downregulated in both BC and liver metastasis. SPARCL1 is a newly discovered player in tumors and is mainly associated with physiological processes, such as cell migration, adhesion, and cell proliferation regulation (Gagliardi, Narayanan & Mortini, 2017). SPARCL1 has been found to be under-expressed in a variety of malignancies and is closely associated with tumorigenesis and metastasis. The expression of SPARCL1 is downregulated in colorectal cancer and is related to tumor differentiation, stage, distant metastasis, and OS (Hu et al., 2012b; Zhang et al., 2022). SPARCL1 is also downregulated in prostate cancer, being associated with disease progression, especially in invasive prostate cancer. Moreover, SPARCL1 can inhibit migration, invasion, and metastasis of prostate cancer (Hurley et al., 2012; Xiang et al., 2013). Similarly, SPARCL1 was found to be a tumor suppressor in gastric cancer (Li et al., 2012), osteosarcoma (Zhao et al., 2018b), pancreatic cancer (Esposito et al., 2007), lung cancer (Deng et al., 2021), and renal cell carcinoma (Ye et al., 2017). SPARCL1 expression in colorectal cancer(CRC) liver metastasis is downregulated, with SPARCL1 being considered the hub gene of liver metastasis and a biomarker with a significant prognostic value (Zhang et al., 2021b). The possible mechanisms underlying the role of SPARCL1 in liver metastasis include the halt of various activities such as platelet activation driving the acquisition of a metastatic signature, participation in the IGFBP-IGF signaling pathway, and promoting the secretion of granules (endocytosis and exocytosis) (Zhang et al., 2021c). The highly consistent expression pattern of SPARCL1 in BC and liver metastasis in CRC may suggest an analogous mechanism for SPARCL1 in BC liver metastasis. Nevertheless, the detail of the mechanism needs to be explored and validated in the future. Furthermore, downregulation of SPARCL1 expression enhances liver metastasis of malignant gastrointestinal stromal tumor cells (Shen et al., 2018). However, few studies have reported on the expression and function of SPARCL1 in BC. Consistent with our findings, SPARCL1 has been reported to be downregulated in BC (Cao et al., 2013). To the best of our knowledge, the expression pattern and role of SPARCL1 in the liver metastasis of BC have not been reported previously. Our research identified SPARCL1 as a hub gene for BC liver metastasis and showed low expression in metastasis tissues. In addition, our clinicopathological correlation analysis revealed that the low expression of SPARCL1 is related to age, TNM stage, ER status, PR status, histological type, molecular type, and survival status of patients with BC, while younger age, higher TNM stage, HR-negative status, and basal-like type were among the unfavorable phenotypes in patients with BC. However, different SAPRCL1 expression levels were not prognostically significant in individual breast cancer subtypes. We suppose that SPARCL1 may primarily influence survival through its involvement in determining breast cancer subtypes.

GSEA data indicated that pathways such as DNA replication, cell cycle, oxidative phosphorylation, and homologous recombination were significantly enriched when SPARCL1 was downregulated, revealing potential molecular mechanisms by which SPARCL1 participates in BC. The most vulnerable cellular process in oncogenic lesions is DNA replication, with oncogene-induced replication stress being a fundamental step and early driver of tumorigenesis (Kotsantis, Petermann & Boulton, 2018). Cell cycle and DNA replication mechanisms are closely coordinated to ensure correct single-genome replication during cell division, thereby avoiding the occurrence of diseases such as cancer (Dai et al., 2021). Homologous recombination repair is an essential pathway of DNA damage repair, and homologous recombination deficiency is considered an important biomarker and risk factor for BC (Den Brok et al., 2017; Shen et al., 2020; Telli et al., 2018).

The in vitro inhibition of BC cell proliferation, migration, and invasion by SPARCL1 indicates the gene acts as a tumor suppressor in BC. However, the specific mechanisms by which SPARCL1 inhibits the proliferation and migration of BC cells warrants additional investigations, and the role of SPARCL1 in BC and liver metastasis needs to be further confirmed. Future experiments are therefore needed to illuminate these critical details. In addition, the transcriptome data in this study were taken from a public database; to obtain more accurate and credible results, the expression and role of hub genes such as SPARCL1 require further experimental verification using liver metastasis tissues and cell lines, such as tissue microarray technology, animal experiments, etc.

Conclusions

In summary, comprehensive screening of DEGs and hub genes revealed that SPARCL1, a tumor suppressor in BC, has the potential to become a diagnostic biomarker and therapeutic target of BC and liver metastasis.

Supplemental Information

Supplemental Information 1 The raw measurements and analysis

SPARCL1 was identified as a hub gene in BC liver metastasis, and expression levels of the SPARCL1 in BC tissues were significantly downregulated compared to normal tissues; SPARCL1 knockdown significantly increased the proliferation and migration of BC cells, whereas the proliferation and migration were suppressed after elevating the expression of SPARCL1. These results were used to determine the critical role of SPARCL1 in BC and liver metastasis.

Click here for additional data file.

Supplemental Information 2 Protein-protein interaction network of the 332 DEGs identified in breast cancer liver metastasis

Each node represents one gene, and the edges between nodes represent interactions. Blue and red indicate down- and up-regulated DEGs, respectively. PPI: protein-protein interaction

Click here for additional data file.

Supplemental Information 3 DEGs of BC liver metastasis identified from GSE58708

(A) volcano plot of the 683 DEGs, including 410 up-regulated and 273 down-regulated genes, blue indicates down-regulated DEGs, red indicates up-regulated DEGs, the cut-off criteria is adj.P.Val < 0.05 and —log.2FC— < 2.0. (B) Heatmap of the DEGs, blue indicates down-regulated DEGs, red indicates up-regulated DEGs. DEGs: differentially expressed genes

Click here for additional data file.

Supplemental Information 4 Validation of the SPARCL1 expression in liver metastasis and BC tissues

(A) The expression of SPARCL1 between liver metastasis and primary tissues in GSE58708. (B) The expression of SPARCL1 between BC tissues and normal tissues in TCGA. Metastasis: liver metastasis of BC, primary: primary tumor of BC, NT: normal tissue, BRCA: breast carcinoma, p.adj: adjusted p-value

Click here for additional data file.

We would like to thank Editage for English language editing.

Additional Information and Declarations

Competing Interests

Author Contributions

Human Ethics

Data Availability

The authors declare there are no competing interests.

Mingkuan Chen conceived and designed the experiments, performed the experiments, analyzed the data, prepared figures and/or tables, authored or reviewed drafts of the article, and approved the final draft.

Wenfang Zheng performed the experiments, analyzed the data, prepared figures and/or tables, and approved the final draft.

Lin Fang conceived and designed the experiments, authored or reviewed drafts of the article, and approved the final draft.

The following information was supplied relating to ethical approvals (i.e., approving body and any reference numbers):

This study was performed in line with the principles of the Declaration of Helsinki. Approval was granted by the Ethics Committee of Shanghai Tenth People’s Hospital (No. 2020-KN174-01).

The following information was supplied regarding data availability:

The raw measurements are available in the Supplemental Files.

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
