# Peer review of "Identifying liver metastasis-related hub genes in breast cancer and characterizing SPARCL1 as a potential prognostic biomarker"

_PeerJ, doi:10.7717/peerj.15311_

## Round 0.1 · original submission · Major Revisions

When assessing your paper, the reviewers identified some issues that must be addressed for the manuscript to be suitable for publication in this journal. In particular, reviewers questioned used of siRNA to knockdown SPARCL1. Instead shRNAs or CRISPR/CAS9 should be used to convincingly show SPARCL1 related biology. Additionally, the authors should overexpress SPARCL1 to validate functional/biological consequences.

Reviewer 1 ·

Basic reporting

No Comments

Experimental design

While this research could potentially be meaningful, in the current format, this manuscript has not clearly been able to fill an identified knowledge gap.
While it is important to identify the mechanisms underlying breast cancer metastasis to the liver, there are already a few manuscripts addressing the role of SPARCL1 in the migration, invasion and metastasis of cancer cells. 23534723, 28927026, 34422629,
To sufficiently address this knowledge gap, the authors should experimentally address a few more critical questions, in addition to discussing these prior findings more thoroughly to describe how this manuscript is adding new findings to the existing body of literature .
Experiementally, in addition to demonstrating reduced expression of SPARCL1 in a transcriptomic database of metastatic samples vs primary samples, it would be important to know if this finding is translated to a protein level. I would request the authors to attempt testing the protein level in paired breast cancer primary and metastatic tissue, perhaps by using a tissue microarray.
Additionally, it would be important to assess if this finding is associated with a specific subset of breast cancer. It would be beneficial if the authors could compare the survival of patients with SPARCL1 high vs low tumors within different breast cancer subtypes.

Validity of the findings

While the experiments described in this study are statistically robust, the siRNA knockdowns used to assess the effect of SPARCL1 are not very convincing. The knockdown achieved by these siRNAs are less than 50% in 2/3 cell lines.
To convince readers, it would probably be more apt if the authors could use a current and more specific technology such as CRISPR to convincingly knockdown this gene and repeat a few critical experiments such as colony formation and invasion assays.

Reviewer 2 ·

Basic reporting

Proof-reading of the manuscript is essential. Introduction lacks sufficient details about the gene studied, SPARCL1.

Experimental design

Research question and hypothesis needs to be better defined. Experimental approach is defined in detail.

Validity of the findings

The knockdown experiments must be conducted in replicates. The knockdown of SPARCL1 appears to be ~30-40%. Authors should consider using different si-RNA or stable knockdowns with sh-RNA in order to demonstrate more drastic biological effect.
The results section needs to be elaborated and proper conclusions for each section should be provided.

Reviewer 3 ·

Basic reporting

No Comment

Experimental design

The experimental evidence that support the bioinformatic analysis is weak and need some detailed experimentation and analysis.
Major concern:
The authors have used siRNA mediated knockdown of SPARCL1 gene in BC cell lines and have observed the effect of downregulation of SPARCL1 on biological functions such as cell viability, proliferation as well as cell migration. Since the expression of SPARCL1 is itself downregulated in BC cells as compared to the normal cells (as already shown by the authors), a better experimental design would be to observe these cell functions in BC cells as compared to the normal cells. That would provide definitive evidence that SPARCL1 is involved in this process.

As a proof of concept and to clearly establish the effect of the downregulation of SPARCL1 in BC cells, it would be good to overexpress SPARCL1 in BC cells and see if the effect of cell viability, proliferation and migration are reversed.

Validity of the findings

No comment

Additional comments

The manuscript entitled “Identifying liver metastasis-related hub genes in breast
cancer and characterizing SPARCL1 as a potential prognostic biomarker” submitted by Chen et al identifies the genes that may regulate liver metastasis during breast cancer. The authors have utilized the publicly available datasets and bioinformatics tools to identify differentially expressed genes and suggests the important role of SPARCL1 during this process.

Overall, the study is planned well. The authors have efficiently mined the data sets and identified the list of genes and the possible pathways which may be regulated. All the bioinformatic analysis is performed extensively. The methods have been described well. However, the experimental evidence that support the bioinformatic analysis is weak and need some detailed experimentation and analysis.

Major concern:
The authors have used siRNA mediated knockdown of SPARCL1 gene in BC cell lines and have observed the effect of downregulation of SPARCL1 on biological functions such as cell viability, proliferation as well as cell migration. Since the expression of SPARCL1 is itself downregulated in BC cells as compared to the normal cells (as already shown by the authors), a better experimental design would be to observe these cell functions in BC cells as compared to the normal cells. That would provide definitive evidence that SPARCL1 is involved in this process.

As a proof of concept and to clearly establish the effect of the downregulation of SPARCL1 in BC cells, it would be good to overexpress SPARCL1 in BC cells and see if the effect of cell viability, proliferation and migration are reversed.

---

## Round 0.2 · accepted · Accept

Thank you for submitting the manuscript.

Reviewer 2 ·

Basic reporting

Authors have addressed my concerns.

Experimental design

Authors have addressed my concerns.

Validity of the findings

Authors have addressed my concerns.

Reviewer 3 ·

Basic reporting

No comment

Experimental design

The authors have addressed the main concern. The over-expression data looks good and support the hypothesis.

Validity of the findings

No comment